# Microscopic Processing of Transparent Material with Nanosecond and Ultrafast Lasers

**DOI:** 10.3390/mi15091101

**Published:** 2024-08-30

**Authors:** Di Song, Jiaqi Wang, Xinyang Wu, Liancong Gao, Jiao Yang, Xiaoxu Liu, Qing Luo, Dongdong Wang, You Wang

**Affiliations:** 1Southwest Institute of Technical Physics, Chengdu 610041, China; disong2022@alu.scu.edu.cn (D.S.); 15100491399@163.com (J.W.); 17803330961@163.com (X.W.); gaolc@jlmu.cn (L.G.); jiaoyang_2019@163.com (J.Y.); lxx8202@163.com (X.L.); lawtsing@outlook.com (Q.L.); mingboor@163.com (D.W.); 2School of Physics and Technology, Xinjiang University, Urumqi 830046, China

**Keywords:** laser manufacturing, ultrafast laser, nanosecond laser, transparent material manufacturing

## Abstract

Due to their excellent light transmission, heat resistance, corrosion resistance, high mechanical strength, and other characteristics, transparent materials have been widely used in emerging industries such as aviation, aerospace, microelectronics, interconnected communication, etc. Compared with the traditional mechanical processing and chemical processing of transparent materials, laser processing, with such characteristics as a high peak power, high energy density, and non-contact processing, has a lot of obvious advantages in processing efficiency and accuracy. In this paper, some of the recent research advancements concerning the laser processing of transparent materials are introduced in detail. Firstly, the basic mechanism of the interaction between the laser and material is briefly summarized on the time scale. The differences in principle between nanosecond, picosecond, and femtosecond pulse laser processing are analyzed. Then, the main technical means of the nanosecond laser processing of transparent materials are summarized. Next, the main application directions of the ultrafast laser processing of transparent materials are discussed, including the preparation of optical waveguide devices, periodic structure devices, micropores, and microchannels. Finally, this paper summarizes the prospects for the future development of laser processing transparent materials.

## 1. Introduction

Due to their excellent light transmittance, mechanical strength, heat resistance, and corrosion resistance characteristics, transparent materials have a wide range of applications in aviation, aerospace, microelectronics, optical communication, optical storage, etc. At present, the main industrial transparent materials include crystalline silica, diamond, optical crystals, amorphous quartz glass, optical glass, and other materials. Due to some features, including a high hardness, high melting point, poor thermal conductivity, and fragility, transparent materials are easy to crack or even break during processing using traditional mechanical and chemical methods. Therefore, it is difficult to meet the requirements of precision processing at the micro–nano scale [1,2,3]. With the development of pulsed laser technology, the improvement of laser beam quality [4], and the emergence of Q-switched technology and mode-locked technology [5], the pulse width of practical lasers has also been greatly shortened from the millisecond level to the current picosecond and femtosecond levels, and the peak power can even reach 10^22^ W/cm^2^, which makes high-quality and high-efficiency laser processing possible for transparent materials.

As a new type of non-contact, high-energy/high-power density manufacturing method, the laser energy can be quickly focused on the surface or inside of the components and can be absorbed by materials in a very short time during laser processing. Compared with the traditional processing methods, the laser processing method has advantages such as little material pollution, higher precision and efficiency, non-contact, etc. Ultrafast lasers have fundamentally changed the mechanism of lasers interacting with matter due to their unique physical properties, which have been given a lot of attention and have developed rapidly in industries. On the time scale, ultrafast lasers inhibit thermal effects, which is conducive to the formation of a higher quality morphology. On the spatial scale, ultrafast lasers can be used to achieve the nano-scale microstructure fabrication and break the optical diffraction limit using a variety of nonlinear processes [6,7]. In recent years, the processing of transparent materials based on nanosecond lasers has received great attention from researchers in both laboratories and industrial lines. The obvious thermal effects can be observed during processing applications using nanosecond lasers and can lead to some processing defects such as chipping and cracks [8,9]. With the optimization of the processing parameters and the use of various auxiliary procedures, the processing quality of transparent materials has been continuously improved using cheap nanosecond lasers. Processing with nanosecond lasers has become a feasible technique for drilling and grinding transparent materials. Compared with other traditional transparent material processing methods, laser processing has become the cutting-edge core technology of transparent material processing with many unique advantages.

So far, many review papers on laser processing of transparent materials have been published [10,11,12,13,14,15,16,17,18,19,20], which basically focus on the processing process and the results of analyses of ultrafast lasers; however, the research results of nanosecond laser processing are rarely involved for large pulse widths. In this paper, the interaction process between pulsed lasers and transparent materials is firstly mentioned, and then the relevant mechanism is expounded for the variation of the physical and chemical properties of transparent materials. Then, the main factors affecting the processing efficiency and quality of nanosecond lasers are introduced when processing transparent materials, and the recent improvement schemes are also referred to in the paper. Afterwards, the research achievements of the preparation of different kinds of devices on transparent materials by using ultrafast lasers in recent years are summarized. Finally, the future development and technical trend of laser transparent material processing methods are discussed, which might provide some suggestions and references for the further development of such processing methods.

## 2. Mechanism of Laser Processing

The process of interaction between lasers and matter can be divided into four parts: carrier excitation, thermalization, carrier removal, and thermal and structural effects, as shown in Figure 1. The interaction between lasers and matter is a very complex process, but the relative mechanism can be summarized as the laser excitation of electrons inside materials. The laser transfers the energy through photons to materials with a high energy/power density in the focused area, followed by a series of related thermodynamic processes. The electrons in the material at the focal point with the highest laser energy/power density are excited to high-energy states after absorbing photons. Due to the generally large bandgap of transparent materials, the energy of a single photon is insufficient to excite electron transitions. At this time, the free electron generation modes are multiphoton ionization and tunneling ionization. If some charge carriers are excited at a rate that is much higher than the bandgap, impact ionization can also generate excited charge carriers. This process takes approximately 10^−13^·s. After being excited, electrons and holes will be redistributed to the conduction and valence bands through carrier–carrier scattering and carrier–phonon scattering. This process will not reduce the number of charge carriers, but the energy of charge carriers will be reduced due to spontaneous phonon emission and transferred to the lattice. This process takes approximately 10^−12^·s. Over time, the charge carriers and lattice will reach a certain equilibrium state. But at this point, the number of charge carriers is higher than that in the thermal equilibrium state, so excess charge carriers will be removed through the recombination of electrons and holes or diffusion from the excitation region. This process takes approximately 10^−11^·s to begin. As energy accumulates, if the lattice energy exceeds its melting and boiling points, melting and gasification will occur. Subsequently, a series of physical processes, such as thermal diffusion, melting, and explosion, as well as photochemical processes, such as phase transition, will occur.

In the whole process of laser–matter interaction, the multiphoton absorption and tunneling ionization play a dominant role in a femtosecond time range. In the picosecond time range, the electrons absorb the photon energy, and then transfer such energy to the crystal lattice through the electron–phonon coupling, and then the energy is finally converted into thermal energy. In the nanosecond time range, the temperature of the material around the laser focus region sharply rises and the pressure waves are generated and rapidly transmitted to the surroundings. In the sub-microsecond time range, thermal energy diffuses along the temperature gradient from the laser focus area, causing the material to melt and micro-explode. The corresponding process is shown in Figure 2.

The time for carriers to reach thermal equilibrium with the lattice and the removal of excess carriers are the key factors that codetermine the laser processing mechanism. Assuming that electrons and lattices are characterized by their temperatures ($T_{e}$ and $T_{i}$), the diffusion of laser deposition energy can be described by a one-dimensional dual temperature model [22]:(1)$$C_{e}\frac{\partial T_{e}}{\partial t}=-\frac{\partial Q\left(z\right)}{\partial z}-\gamma \left(T_{e}-T_{i}\right)+S,$$
(2)$$C_{i}\frac{\partial T_{i}}{\partial t}=\gamma \left(T_{e}-T_{i}\right),$$
(3)$$\left(z\right)=-k_{e}\partial T_{e}/\partial z,\;S=I\left(t\right)A\alpha exp\left(-\alpha z\right).$$

Among them, $C_{e}$ and $C_{i}$ represent the heat capacity of the electronic and lattice systems, respectively, $Q\left(z\right)$ is the heat flux density, γ is the parameter characterizing the electron lattice coupling, S is the laser heating source term, $k_{e}$ is the electronic thermal conductivity, A is the surface transmittance, and α is the material absorption coefficient. Formulas (1)–(3) include three characteristic time scales: electron cooling time $\tau_{e}(\tau_{e}=C_{e}∕\gamma$), lattice heating time $\tau_{i}(\tau_{i}=C_{i}∕\gamma$), and laser pulse duration $\tau_{L}$. Therefore, the material removal mechanism corresponding to different laser pulse widths will also vary.

The material removal mechanism for ultrafast pulsed lasers has been researched for many years [23,24,25]. The material removal mechanism of ultrafast pulsed lasers is explained as follows. Due to the ability of ultrafast lasers to inject energy into materials within a time range of picoseconds or even femtoseconds, the electric field force exerted by lasers on electrons in materials can be compared to the Coulomb force of atomic nuclei, leading to multiphoton ionization. At the same time, the deformation of the valence band electron Coulomb field forms a potential barrier in the polarization direction of the laser, resulting in tunneling ionization. As the number of free electrons generated by various nonlinear processes increases, collision ionization and free carrier absorption processes continue, ultimately leading to an avalanche-like increase in the free electron concentration. During this process, the duration of the laser pulse, $\tau_{L}$, is less than the electron cooling time, $\;\tau_{e}$. Equation (1) can be rewritten to obtain the electron temperature after the laser pulse [22]:(4)$$T_{e}\left(\tau_{L}\right)\approx {\left(\frac{2F_{a}\alpha }{C_{e}^{\prime }}\right)}^{\frac{1}{2}}\exp\left(-z\alpha /2\right),$$

Among them, $F_{a}=I_{0}A\tau_{L}$ represents the absorbed laser energy, and $I_{0}$ is assumed to be a constant. The initial conditions for electron and lattice temperatures can be obtained from Equation (4). The temperature of the electrons which are excited to high-energy states is much higher than the temperature of their lattice, breaking the original electron lattice equilibrium. The temperature that the lattice can reach is determined by the average cooling time of the electrons:(5)$$T_{i}\approx T_{e}^{2}\left(\tau_{L}\right)\frac{c_{e}^{\prime }}{2c_{i}}\approx \frac{F_{a}\alpha }{c_{i}}\exp\left(-\alpha z\right),$$

Significant material evaporation can occur when $C_{i}T_{i}>\rho \Omega$, where ρ is the material density and Ω is the specific heat of evaporation. According to Equation (5), the strong ablation conditions under the action of an ultrafast pulse laser can be obtained as follows:(6)$$F_{a}\geq F_{th}\exp\left(\alpha z\right),$$

Among them, $F_{a}$ is equivalent to ρΩ/α, representing the evaporation threshold of the material under ultrafast laser action. As energy is released into the lattice, the lattice bonds of the material break. This process occurs at the picosecond scale, and the material is not melted or vaporized in time and is directly ejected before nucleation in the liquid and gas regions.

The duration of a single pulse of a nanosecond pulse laser on materials is mostly in the tens or hundreds of nanoseconds. Therefore, the mechanism of material removal by a nanosecond pulse laser should not only include various effects in the femtosecond and picosecond time ranges, but also consider microprocesses at the level of hundreds of nanoseconds or even microseconds. In this case, the temperature of the electron and the lattice are the same ($T_{e}=T_{i}=T$), and Equations (1)–(3) can be simplified as follows:(7)$$C_{i}\partial T∕\partial t=\partial ∕\partial z\left(k_{0}\partial T∕\partial z\right)+I_{a}\alpha \exp\left(-\alpha z\right)$$

The main loss of energy in the interaction between the nanosecond laser and matter is the thermal conduction in the object. The laser intensity and flux under strong ablation conditions can be written as follows [22]:(8)$$I>I_{th}\sim \frac{\rho \Omega D^{1∕2}}{{\tau_{L}}^{1∕2}},\;F>F_{th}\sim \;\rho \Omega D^{1∕2}\times {\tau_{L}}^{1∕2}.$$

Among them, $D=k_{0}/C_{i}$ represents the thermal diffusion coefficient, and $k_{0}$ is the conventional equilibrium thermal conductivity of the material. After the transfer of electronic energy to the lattice, the material lattice fractures, causing the material to expand. A large number of charge carriers continue to act on the lattice, directly melting and vaporizing the material. The sharp increase in local temperature leads to a high temperature difference at the processing edge, forming local micro-explosions and producing microcracks or even faults. The resulting thermal stress further causes the material to fracture and detach. The melted and vaporized material will solidify below its melting and boiling points and reattach to the material.

It is difficult to remove the thermal influence for processing by nanosecond laser pulses in principle, and the obvious deposition and recast layers can be observed in this case. Additionally, there are also some obvious modified regions around the processing area, making it difficult to be applied in the fields of complex micro–nano processing. However, because of the low price and high average power, the nanosecond pulse lasers are welcomed in the fields of drilling, surface etching, and cutting with the requirement of relatively lower precision. Using femtosecond pulse lasers, the formation of heat zones can be suppressed in principle and the optical diffraction limit can be broken through various nonlinear effects, making them widely used in the fabrication of various micro–nano devices. However, the actual laser processing method is not a single pulse processing method, but involves processing with a series of pulses acting together, so it is still affected by thermal effects, even with the utilization of an ultrafast laser. At the same time, the average power of an ultrafast laser is relatively low, and the time repetition rate is also difficult to raise, so improving the processing efficiency should be a daunting task for ultrafast lasers. 

Figure 3 shows schematic diagrams of processing transparent materials using a nanosecond pulse laser, picosecond pulse laser, and femtosecond pulse laser, respectively. The basic processing equipment consists of a laser light source, an optical path control system, and a scanning path controller.

The single pulse energy of a nanosecond pulse laser is usually in the mJ range, with a repetition rate of several hundred kHz. In practical processing applications, the thermal effect is obvious, the laser modification range is large, and the processing accuracy is low. In order to reduce the influence of thermal effects, laser sources in the ultraviolet band are usually chosen. Meanwhile, to reduce the influence of thermal stress, wet etching is commonly used when processing with nanosecond pulse lasers. Wet etching reduces the risk of material edge breakage and cracking. At the same time, by selecting appropriate excitation solutions, chemical reactions can be used to assist laser etching, and the laser wavelength can be extended to near infrared lasers, freeing the limitation of the light source wavelength. The efficiency of nanosecond pulse laser processing is very high, and the etching scale is usually at the millimeter level. A picosecond laser significantly reduces the thermal effects during the processing by reducing the pulse width. However, the heat accumulation during the processing is still the main factor affecting the processing quality. The single pulse energy of a picosecond laser can also reach the mJ level, and the pulse frequency is also in the hundreds of kHz range. Similar to nanosecond pulse laser processing, picosecond pulse laser processing usually involves the selection of short wavelength light sources and auxiliary methods such as air cooling and increasing the water film to reduce the thermal stress of materials in order to reduce the influence of thermal effects. The single pulse energy of a femtosecond pulse laser is relatively low, usually at the μJ scale, with a low repetition rate, usually in the range of several kHz, and the etching scale is mostly at the μm level. The extremely high peak power allows femtosecond pulse lasers to be unrestricted by the laser wavelength, and the selection range of light sources is much larger than nanosecond and picosecond pulse lasers. At the same time, the characteristic of cold ablation makes the machining morphology more controllable and the machining size smaller. Due to the characteristics of femtosecond pulse lasers, traditional Gaussian beam spatial shaping can be achieved by absorbers, beam integrators, and field mapping to generate pulses with specific and special spatial shapes such as flat top beams, Bessel beams, and Airy beams. This significantly improves the reduction in the hole taper and enhances the quality of the processing morphology. At the same time, femtosecond pulse lasers can be time domain-shaped, greatly improving the depth to the diameter ratio of processed holes. These studies are expected to be applied in the preparation of special microfluidic, optical waveguide, and other devices.

## 3. Processing of Transparent Materials with Nanosecond Lasers

At present, research on processing transparent materials with nanosecond lasers mainly focuses on application fields such as drilling, cutting [29,30,31], preparing microchannels, etc. [32]. Due to the inherent characteristics of nanosecond lasers, the thermal zone is significant during the interaction between lasers and materials [33,34,35]. The instantaneous high temperatures can cause a significant temperature gradient around the laser focusing area. The induced significant thermal stress will lead to problems such as edge collapses, cracks, and scum during the processing. Generally speaking, to improve the processing quality for nanosecond lasers, the most important thing is to suppress the influence of the hot zone on the processing.

As early as 1996, C. Koerner and R. Mayerhofer et al. conducted processing experiments on transparent materials using a copper vapor nanosecond laser [36]. They undertook drilling experiments on phenolic resin and quartz glass with a copper vapor nanosecond laser with a wavelength of 578 nm and pulse width of 50 ns, respectively. They summarized four basic processes related to the melting mechanism, including the following: (1) the preliminary cracking of materials; (2) the melting of materials near the evaporation threshold; (3) static evaporation and melting; (4) melt displacement and injection. At the same time, these authors have also focused on the absorption properties of materials. For brittle materials, it is necessary to strictly classify strong and weak absorbers, and different laser powers should be adopted for different types of materials to simultaneously achieve the best processing quality and efficiency.

To reduce hot zones during laser processing, in 2002, H. Niino et al. used a nanosecond-pulsed KrF excimer ultraviolet laser with a wavelength of 248 nm to excite the pure toluene solution and performed laser-induced backside wet etching (LIBWE) on silica glass plates [37]. Materials are directly ablated during the traditional laser processing method with an air–solid processing interface. The nonlinear effects and the influence of the heat zone make laser processing very difficult on hard and brittle materials. To overcome these difficulties, the authors changed the processing interface to a liquid–solid interface. The results revealed that the heat-affected zone and the effects of debris accumulation on the processing have been successfully reduced. At the same time, they also focused the laser through the transparent material on the liquid–solid interface and found that the absorption of laser energy had been reduced by liquid and the material had been pre-heated. By means of this method, the thermal stress in materials can be reduced and thereby the generation of debris and cracks can be suppressed. As shown in Figure 4, they produced a clear grid micropattern with no debris or microcracks around the etched area. In the experiment, they successfully created different microchannels inside transparent materials such as silica glass, quartz, calcium fluoride, and fluorocarbon resin. In their subsequent research, they increased the etching rate while reducing the laser flux, with no generation of cracks and fragments, which is significant in promoting the scale and mass production of transparent materials processed by lasers.

In 2002, X. Ding et al. [38] and R. Böhme et al. [39] used nanosecond lasers to complete high-quality wet etching on quartz materials. In the following year, K. Zimmer et al. used nanosecond lasers to accomplish high-quality processing on glass [40]. These technologies can be used for the high-quality surface processing of transparent materials without any cracks. In 2005, J.Y. Cheng et al. used a UV LD-pumped solid-state laser with a high repetition rate of 6 kHz and a pulse width of 266 nm to perform LIBWE processing on glass in microfluidic chip manufacturing. They achieved crack-free direct writing on glass, and created grooves that are 100 μm wide, 10 μm deep, and 100 mm long within 12 min [41]. In 2013, D. Nieto et al. prepared microchannels on sodium calcium glass using an Nd:YVO_4_ laser [42] and obtained microchannels with a minimum diameter of 8 mm and a depth of 1.5 mm, respectively. They conducted a thermal reflux treatment on the sample after laser etching to eliminate the glass debris generated by laser etching. They reduced the average roughness generated by laser ablation by two orders of magnitude, reaching the roughness level of the glass material before processing. In 2015, Jiajun Shen et al. used a 532 nm nanosecond laser (produced by EdgeWave GmbH of Germany) to conduct a glass cutting experiment by changing the processing parameters of the laser [43]. They focused on the influence of the laser process parameters on the glass cutting effect. The experimental results have shown that by optimizing laser parameters such as current size, filling interval, layered cutting, and cutting speed, they can process glass substrates with smooth edges, no lateral microcracks, and no debris, and achieve a good product yield. In the same year, Yan Hu et al. used a 532 nm green nanosecond laser to explore the basic rules of laser etching in quartz glass and the effects by changing four parameters, including the laser energy density, defocusing amount, scanning times, and scanning speed, during the process [44]. The processing results are shown in Figure 5.

One of the main parameters affecting the etching depth is the laser energy density. The etching depth is approximately linear with the logarithm of the laser energy density. It is important to note that when the laser power is much smaller than the material burn threshold, the laser cannot cause any damage to the glass target. In addition, the defocus factor is also one of the key factors in laser transparent material processing. When the defocus value is 0, the etching depth is maximized, and both positive and negative defocusing will reduce the etching depth. At the same time, when the scanning frequency is small, the increase in scanning frequency will slow down the etching process and reach an approximate flat trend for glass targets. When the scanning frequency is large, excessive heat accumulation will increase the thermal stress, leading to the generation of cracks and over melting for glass. In fact, the scanning speed is also one of the key parameters in the laser processing of transparent materials. When the scanning speed is too high, the thermal deposition on the scanning path becomes relatively less high, and the removal rate becomes low too. If the scanning speed is too low, additional heat will be accumulated and the etching quality will be reduced. Yan Hu et al. also studied the quality of laser processing on quartz in three types of environments, including hot water, cold water, and compressed air environments. They also studied the laser processing quality with different processing starting surfaces. The results show that the processing quality in three environments is better than that of direct processing in air. At the same time, the processing quality when the focal point is moved from the bottom to top in glass is significantly higher than that when the focal point us moved from the top to bottom, which should be due to the pre-heating of the removal area by the laser beam.

In 2020, Yong Jiang et al. focused on the laser-induced damage to BK7 (K9) glass components and used the online imaging technology to obtain side and front images of the laser-induced damage. They analyzed the characteristics of the filamentation damage and the body damage and the effects of nucleation damage cracks on the damage growth [45]. In the same year, W.Y. Li et al. used a 532 nm nanosecond pulsed laser to cut the solar float glass with a thickness of 2.5 mm [46]. They replanned the cutting path using the Hybrid Bottom-up Multilayer Increment and the Spiral Line (HBMISL) method, as shown in Figure 6.

These authors used a method named the Hybrid Bottom-up Multilayer Increment and the Spiral Line (HBMIS) method to examine how the parameters, such as the scanning speed, laser pulse repetition rate, helix width, and helix overlap rate, affect the perforation results. They designed and conducted 37 sets of single factor cutting experiments to explore the effects of surface quality and cutting efficiency. The results show that when the laser pulse repetition frequency is 55 kHz, the corresponding laser power is 22.6 W with a single pulse energy of 0.41 mJ, the scanning speed is 300 mm/s, and the spiral trajectory parameters are 0.45 mm and 70%, respectively. Further, the minimum cutting width becomes 104.81 μm and the cutting removal rate becomes 4.712 mm^3^/s, respectively. In 2021, N Abbasi adopted the technology of laser-induced plasma-assisted ablation (LIPAA) to perform deep hole drilling on glass by using an Nd:YAG nanosecond pulse laser with a pulse width of 12 ns as the laser source and copper as the metal target [47]. They studied the microdrilling of glass under different kinds of laser intensities. The method they used in the experiment is low-cost and fast and can be used for high-speed microdrilling with both a high-quality and high aspect ratio without significant surface damages to their transparent material. In 2024, Y. Chen et al. studied the surface damage induced by micropores in transparent ceramics under nanosecond laser irradiation [48]. They utilized the annealing process to effectively manage the density and size of micropores, establishing a correlation between micropores in relation to damage thresholds. In the same year, S. Biswas et al. ablated the PMMA plate by using an Nd:YAG nanosecond pulsed laser with the laser power of around 9 W, pulse frequency of around 33 kHz, and cutting speed of 2 mm/s [49]. They successfully made microchannels with an adequate depth on a thick transparent PMMA plate by using the transmission cutting technique.

In addition to LIBWE, HBMIS, and LIPAA, there are other laser processing technologies such as Laser Ablation in Liquid (LAL) and Etching Assisted Laser Modification (EALM). Table 1 summarizes five processing methods for the nanosecond laser processing of transparent materials.

Compared to picosecond and femtosecond lasers, nanosecond lasers are much cheaper, and the relative processing efficiency is higher, but the thermal effect seriously limits their practicality. Whether it is a matter of extending the wavelength of nanosecond lasers to ultraviolet wavelengths, changing the processing interface environment, performing thermal reflux treatments, changing the processing combination parameters, or varying the laser scanning path, researchers want to reduce the influence of the thermal zone generated by the interaction between the nanosecond lasers and the target. At present, the research on transparent material processing using nanosecond lasers mainly focuses on exploring the influence of thermal accumulation on processing effects and selecting different combinations of parameters. Through these studies, the higher processing quality (including low roughness, low cracking, low edge collapse, low taper, etc.), higher processing efficiency, and lower processing costs can be achieved for various transparent materials. At the same time, the processing of transparent materials using nanosecond lasers has shown a “de-auxiliary” trend, which means many assistant procedures become unnecessary during laser processing. The question of how to reduce the dependence on auxiliary technologies during processing is becoming one of the main research topics for research departments and industries.

## 4. Processing of Transparent Materials with Ultrafast Lasers

The pulse duration of an ultrafast laser is very short, and the thermal diffusion scale is much smaller than the depth of the laser penetration. Materials can be ionized by ultrafast lasers at an extremely high speed, and material vapors and nano-scale debris will be directly formed. During the entire laser process, because the energy at higher electron temperatures is not transferred to the material lattice in time, the material melting and thermal diffusion processes during processing can be ignored. In addition, there are many nonlinear effects during processing with ultrafast lasers, which can be used to break the optical diffraction limit and reduce the processing scale to the nanometer level. Until now, ultrafast lasers have been widely used in the preparation of optical waveguide devices [50,51,52], periodic structure devices [53,54,55], micropores, microchannels [56,57,58], etc.

Materials with a low refractive index are actually used to form the core of a conventional optical waveguide wrapped with materials with a high refractive index. When light waves are emitted from a dense medium to a sparse medium, the total reflection will occur while the incident angle is greater than the certain critical angle. Such a theory constitutes the working mechanism of any optical waveguide devices. Using ultrafast lasers to irradiate transparent materials, the refractive index within the irradiation area will change, allowing light waves to be confined to a tiny region at the micrometer level. At present, the physical explanations for the variations in the refractive index of transparent materials induced by ultrafast lasers mainly include the theories of melting–remelting and color center induction. According to the different refractive indices of the working area, optical waveguides directly written by a femtosecond laser are mainly divided into Type I and Type II waveguides. Type I waveguides refer to using a femtosecond laser to increase the refractive index of the irradiation area. Type II waveguides refer to using a femtosecond laser to write two closely spaced straight lines with a reduced refractive index, forming the middle region as a waveguide region. As shown in Figure 7, the area enclosed by the dashed line is the light guiding area [59].

In 2004, L. Gui et al. first engraved optical waveguides in lithium niobate crystals by using femtosecond lasers [60]. In 2006, X. Wu et al. used a femtosecond laser to directly write optical waveguides in periodically polarized lithium niobate crystals, and they successfully generated second harmonics in the waveguides [61]. In 2008, G.D. Marshall et al. used a femtosecond laser with a repetition rate of 1 kHz to directly write optical waveguides and Bragg gratings in Er/Yb co-doped phosphate glass and achieved the so-called waveguide lasers [62]. In the same year, E. Mazur et al. used femtosecond lasers to process polymethyl methacrylate and successfully realized a waveguide structure for single-mode transmission at a wavelength of 632.8 nm [63]. In 2011, Kazuyuki Hirao et al. used a sapphire femtosecond laser to directly write waveguides in LiTaO_3_ crystals, and they investigated the effects of laser energy and pulse duration on the changes in the induced structural refractive index. They summarized the relationship between the waveguide polarization selectivity and the stress distribution around the laser excitation region [64]. In 2015, V. Guarepi et al. utilized writing technology with a femtosecond laser and used a dual-line writing method to design and fabricate linear and curved guiding structures for optical waveguides [65]. In these structures, all the optical wave losses are less than 2 dB/cm. In the same year, S. Kroesen et al. prepared a quasi-phase matched waveguide structure by using the laser direct writing method and generated second harmonics. From their results, using the laser-induced quasi-phase-matched grating with a length of 6 mm, they achieved a maximum conversion efficiency of 5.72% [66]. In 2022, B. Wu et al. used femtosecond lasers to write Type I and Type II waveguides into lithium tantalate crystals (LiTaO_3_) to fabricate an integrated 1 × 5 beam splitter [67], as shown in Figure 8 and Figure 9. The composite orbital cladding waveguides composed of Type II waveguide structures were used for optical signal transmission, reducing photon crosstalk and mode field modulation. The single line waveguides composed of Type I waveguide structures were used for beam splitting. Due to the relatively weak thermal stability of Type I waveguides with single lines, the waveguides can be used to fabricate recoverable and rewritable beam splitters, and a good transmission quality can also be obtained for the rewritten structure. This provides a practical method for the development of erasable photon data processors.

In 2022, B.S. Sun et al. prepared fiber compatible glass waveguides based on the femtosecond laser [68], namely, spherical phase-induced multi-core waveguides (SPIM-WG). The accurate deformation of the cross-section can be achieved along such a waveguide, and the high-resolution shapes and sizes can be finely controlled along both the horizontal and vertical directions. These waveguides have a high refractive index difference of 0.017, low propagation loss of 0.14 dB/cm, and extremely low coupling loss of 0.19 dB, respectively. These waveguides can operate in the ultra-wideband spectral domain, covering wavelengths from the visible region to the infrared region. At the same time, these waveguides have paved the way for the industrialization of packaging and integration of photonic-integrated circuit devices using optical fibers for input and output.

By controlling parameters such as the laser power, femtosecond lasers can also be used to prepare periodic microstructures inside [69] and on the surface of transparent materials. As early as 1965, M. Birnbaum et al. used linearly polarized long pulse lasers to irradiate semiconductor materials and observed a series of periodic stripes on their surfaces [70], named Laser-induced Periodic Surface Structures (LIPSSs). When the energy density of the laser pulse is close to the damage threshold of the material, a series of periodic ripples will be generated on the surface of the material. In 2018, S.H. Messaddeq et al. used a femtosecond titanium sapphire laser with a repetition rate of 1 kHz, pulse width of 100 fs, and frequency of 800 nm, respectively, to ablate As_2_S_3_ chalcogenide glass and induce the LIPSS on the material surface [71]. By controlling the irradiation parameters of the femtosecond laser, the high spatial frequency LIPSS ripples, which are parallel to the polarization of the incident beam, were formed. Nano gaps also appeared, with an average diameter of about 300 nm and a depth of 200 nm between ripples. In addition, the experimental results also showed that complex ripple structures were formed during the transition interval between the high spatial frequency LIPSS features and low spatial frequency LIPSS features. These ripple structures are grid-like and parallel to the polarization direction of the incident light, which is referred to as the cross-superimposed LIPSS.

This cross-overlapping LIPSS can be obtained by ablating chalcogenide glass with appropriate laser parameters. Such LIPSSs have never been observed during the laser processing of other types of glass doped with oxides, phosphates, or fluorides. The formation of a grid-like cross-stacked LIPSS in As_2_S_3_ chalcogenide glass is considered to be mainly related to two factors. The first factor is the interaction between the incident laser beam and surface plasmon polariton waves. The second factor is the self-organizing effect generated by heat accumulation. In addition, the direction of the ripple strongly depends on the polarization direction of the incident beams. These novel nanostructures may be suitable for preparing some infrared-integrated optical devices. In 2018, S. Schwarz et al. used femtosecond pulse lasers to irradiate fused silica and sapphire; they also prepared uniform low spatial frequency LIPSSs [72]. At the same time, they studied the periodic morphology quality of LIPPSs by changing parameters such as the power density and incidence angle of the pulsed lasers.

Ultrafast lasers also have a wide range of applications for the manufacturing of micropores and microchannels in transparent materials. The purpose is to improve the processing efficiency while reducing or eliminating phenomena such as cracks, edge collapses, tapers, etc.

In 2015, Songling Xing et al. selected quartz glass as the experimental material and studied the effect of femtosecond laser parameters on the depth to diameter ratio and morphology of the micropores [73]. The results indicate that the energy of femtosecond laser pulses and the drilling speed have a significant impact on the depth to diameter ratio of micropores. At the same time, they also analyzed the common defects in micro hole processing using femtosecond lasers. In the same year, H. Hidai et al. investigated the fabrication process of holes with a high depth to diameter ratio on glass by using an ultraviolet laser and conducted numerical and experimental analyses of the heat accumulation during the laser preparation. They also investigated the relationship between the melting threshold and the pulse repetition frequency [74]. In 2018, J. Shin also utilized a UV picosecond laser to investigate the effects of three parameters, such as the laser power, scanning speed, and scanning frequency, on the processing quality [75]. The relevant experimental setup is shown in Figure 10.

In 2016, Pengyu Yin et al. conducted a systematic experimental study on parameters such as the pulse energy density, wavelength, spot coupling rate, processing frequency, repetition rate, filling spacing, and focusing position [76]. These authors explored the machining quality and efficiency through the spiral ring cutting method and water surface infiltration machining method, in terms of the scanning path optimization and machining environment change. Figure 11 shows a schematic illustration of the scanning path using the spiral ring cutting method, and Figure 12 is a schematic illustration of processing with the water surface infiltration method.

The experimental results show that, under the fixed laser parameters and processing time, compared with the traditional vertical scanning path method, the energy density in the spiral ring cutting method is higher, making it easier to obtain a larger etching volume. Not only is the processing efficiency much higher than that of the traditional vertical scanning path method, but the processing taper can also be significantly reduced. One of the difficulties for the water surface infiltration processing method concerns how to reduce the energy loss caused by the laser absorption in water. By utilizing the naturally generated eddy currents in water, the accumulated residue in the etching grooves can be cleaned up. In addition, due to the cooling effect of water, it can also reduce the secondary erosion phenomenon that is prone to occur on the lower surface, and the surface quality obtained from processing is significantly improved. The taper of the inner wall of the machining groove can be reduced to a certain extent.

In 2018, Y. Berg et al. utilized a femtosecond infrared laser with a large spot size to achieve the drilling with a depth to diameter ratio of 10:1 in the glass target with a thickness of 1 mm [77]. This process relies on the balance between the nonlinear Kerr effect and the multiphoton absorption in glass; using a large laser spot size is also beneficial for the high-speed processing of thick glass. In the same year, E. Markauska et al. used a picosecond laser with a repetition rate of 100 kHz and a wavelength of 532 nm to ablate sodium calcium glass plates. By adding a thin water film on the glass surface, the ablation efficiency was successfully increased by 12 times [27].

In Table 2, four kinds of transparent material manufacturing applications using ultrafast lasers are compared and summarized.

In order to simulate the interaction between femtosecond lasers and matter more accurately, scholars have carried out in-depth research on the whole micromachining process of femtosecond lasers. In 2018, K. Bergner et al. conducted spatiotemporal analyses on glass processing by using ultrafast pulse lasers. They adopted a time-resolved microscope to obtain the shadow and interference images of the ablation process to study the spatiotemporal evolution and energy transfer of free charge carriers. Meanwhile, they also explored the permanent changes in the glass volume by using femtosecond and picosecond pulsed lasers [78]. In 2021, H. Jo et al. applied high-speed cameras to the pump–probe imaging method and studied the relationship between the high-speed phenomena and the damage occurrence during the machining process [79]. They used the imaging system to observe ultrafast phenomena on a time scale from picoseconds to nanoseconds, as well as slow phenomena on the millisecond time scale, while maintaining a high spatial resolution. The results indicate that the processing damage is mainly generated by the propagation of stress waves. In 2023, A.M. Dikandé et al. proposed a generic model to describe the dynamics of inscription with femtosecond lasers in transparent materials characterized by saturable nonlinearity [80]. In 2024, I.N. Ndifon et al. examined the dynamics of using femtosecond lasers in transparent materials with non-Kerr nonlinearity [81]. The authors proposed a model to explain the effects of the competition between the electron-hole radiative recombination and single-electron diffusion processes on the spatiotemporal profiles of the propagating optical field and the plasma density. In the same year, Y.P. Song et al. investigated and discussed the strong energy-dependent correlation between the evolution of the micro/nano-scale morphology and stress state in laser-induced structures [82].

## 5. Conclusions

Recently, the laser processing of transparent materials has become a hot topic in the field of laser processing. On the one hand, the demand for micro–nano processing of transparent materials in many fields such as microelectronics, biomedicine, and aerospace are increasing sharply. On the other hand, lasers have been dramatically developed and widely applied in recent decades. The emergence of a large number of new lasers with shorter pulse widths, a higher beam quality, more types, and lower costs has greatly promoted the development of theories and experiments concerning transparent material processing using lasers. According to the different mechanisms of material removal, there are some significant differences between the nanosecond laser processing of transparent materials and the ultrafast laser processing of transparent materials in the relative research results. In Table 3, the typical processing results of transparent materials processed by nanosecond, picosecond, and femtosecond pulse lasers are listed.

Nanosecond pulse lasers are cheap and have a high average power, which result in great application advantages in industrial drilling and etching. However, the theoretical research shows that nanosecond pulse laser cannot avoid the influence of thermal effects, and the processing quality is obviously limited. In terms of improving the processing quality, the current focus is on the processing process and post-processing. Wet etching and changing the scanning path are often used to reduce the thermal stress of materials in the processing process. After processing, thermal reflux treatments and electrochemical treatments are often used to reduce the roughness of the processing interface and improve the quality of the morphology. At present, nanosecond laser pulses are usually Gaussian distribution pulses. By changing the temporal and spatial distribution of the laser pulse energy, it may be possible to effectively improve the energy deposition of materials in the processing process, improve the quality of nanosecond pulse laser processing, and expand the scope of application.

Although the research on transparent material processing with ultrafast lasers is more recent than research on nanosecond lasers, the former has attracted significant attention by many scholars due to their unique characteristics. There are many nonlinear phenomena in the interaction between ultrafast lasers and matter, which may break the optical diffraction limit at the theoretical level and reduce the processing scale to the nanometer level. Due to the “cold processing” characteristics of ultrafast lasers, various complex manufacturing processes for micro and nano components can be achieved. Ultrafast lasers have been widely used in the processing of small optical components, microfluidic devices, multifunctional structural surfaces, and other devices. However, ultrafast lasers have problems such as increases in the repetition frequency and average power, resulting in a low efficiency and smaller scale in actual processing applications. These issues indeed limit the industrial applications of ultrafast lasers in the field of transparent material processing. With the in-depth research on non-diffraction lasers, spatial pulse-shaping, GHz technology, time-domain pulse-shaping, spatio-temporal focusing technology, and other cutting-edge technologies, various problems in the processing process have been gradually improved, and the application range of the ultrafast laser processing of transparent materials has been expanded; as a result, achieving the preparation of low SNR optical waveguides, high aspect ratio channels, complex microfluidic structures, and other devices is to be expected.

In addition, researchers in the laser industry have carried out a lot of microscopic studies on the interaction between lasers and transparent materials, and have discovered many interesting phenomena, such as the laser filamentation effect and the ablation-cooling effect, which contribute to the deep understanding of the processing mechanism. It is very likely that some subversive methods and processes for processing transparent materials will emerge at the principal level in the coming years. With the deepening of research, the processing of various special needs will be rapidly progressed, and will play a greater role in biomedical, aerospace, communications, energy security, and other application fields.

## Figures and Tables

**Figure 1 micromachines-15-01101-f001:**
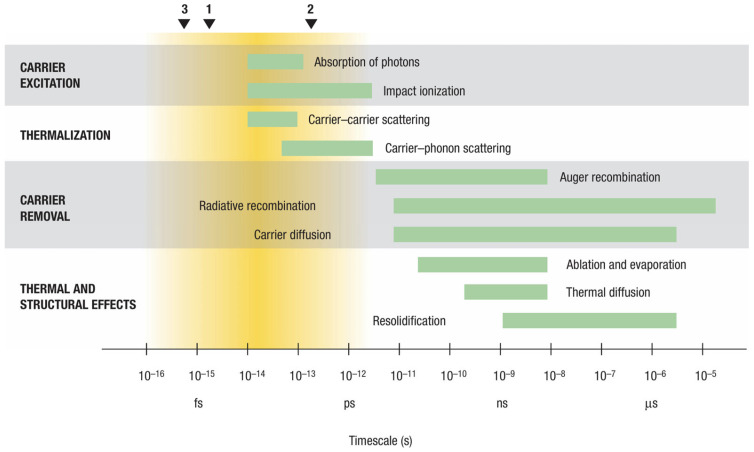
Process of interaction between femtosecond lasers and solid-state materials [21].

**Figure 2 micromachines-15-01101-f002:**
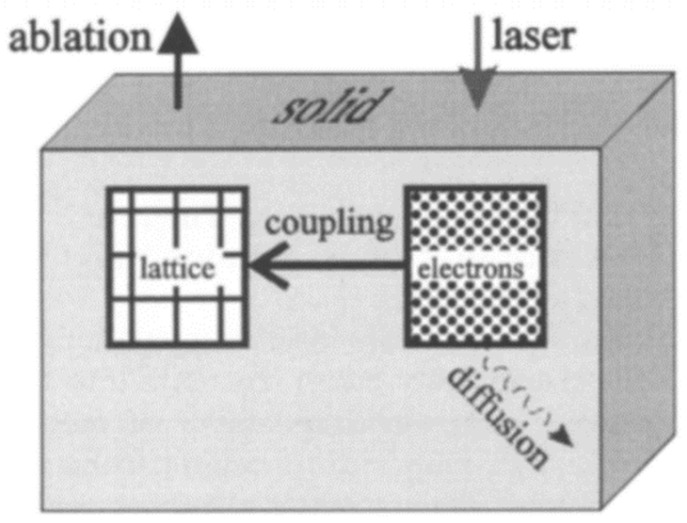
Schematic diagram of the interaction between the laser and the matter [22].

**Figure 3 micromachines-15-01101-f003:**
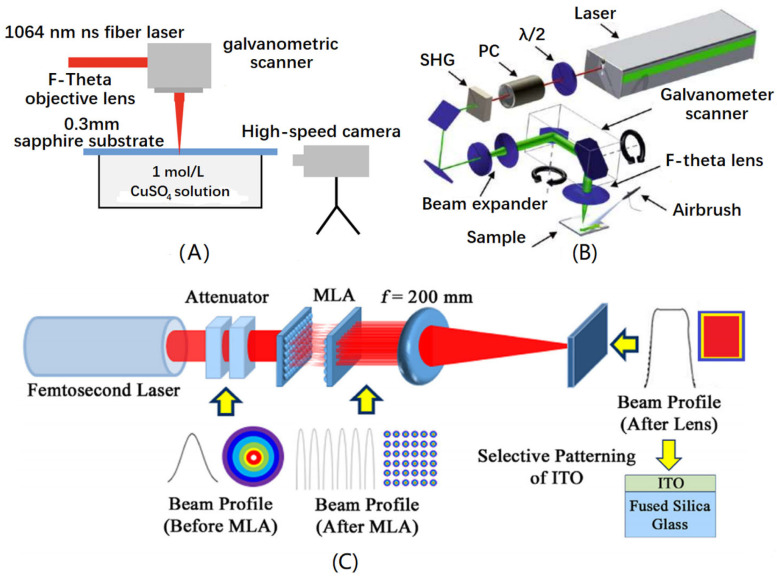
Schematic diagram of laser pulse processing. (**A**) Nanosecond wet etching [26]. (**B**) Picosecond processing [27]. (**C**) Spatially shaped femtosecond laser processing [28].

**Figure 4 micromachines-15-01101-f004:**
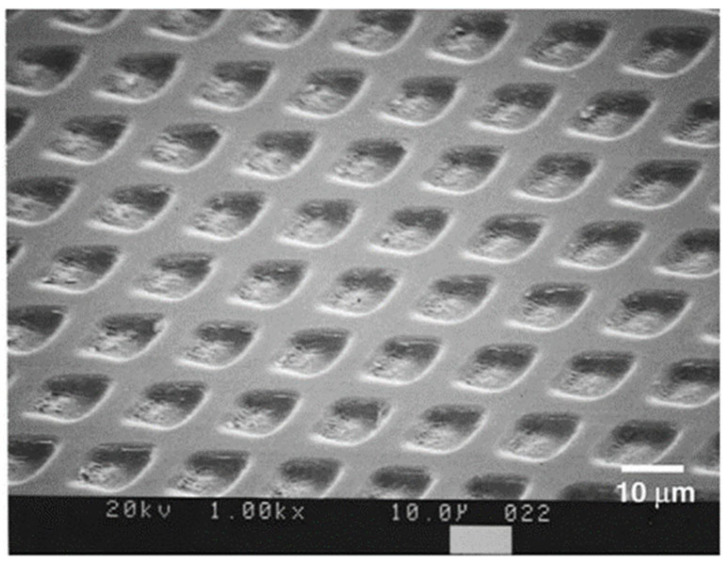
SEM image of grid patterns on the surface of silica glass [37].

**Figure 5 micromachines-15-01101-f005:**
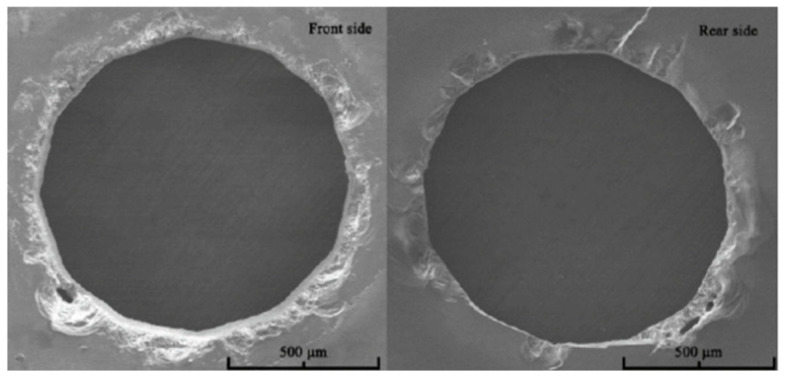
Front and back morphologies of a through-hole [44].

**Figure 6 micromachines-15-01101-f006:**
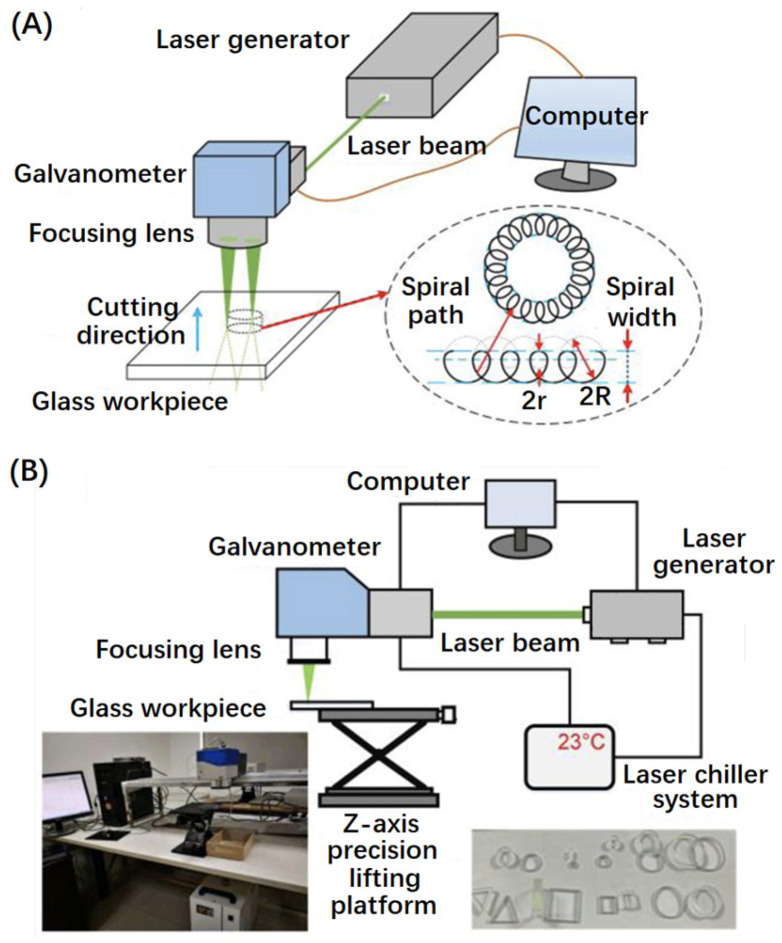
Hybrid Bottom-up Multilayer Increment and the Spiral Line method. (**A**) Schematic illustration of the laser drilling and cutting. (**B**) Experimental setup [46].

**Figure 7 micromachines-15-01101-f007:**
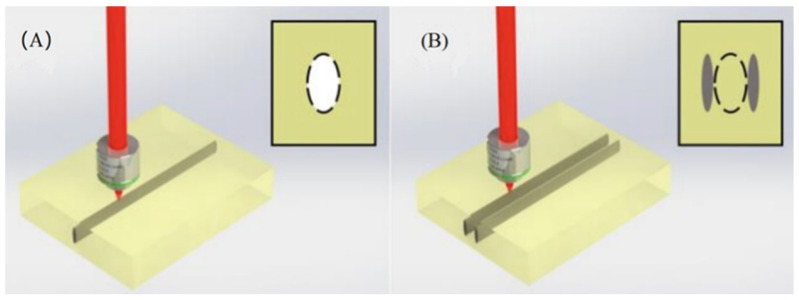
Schematic illustration of the optical waveguides. (**A**)Type I waveguide. (**B**) Type II waveguide [59].

**Figure 8 micromachines-15-01101-f008:**
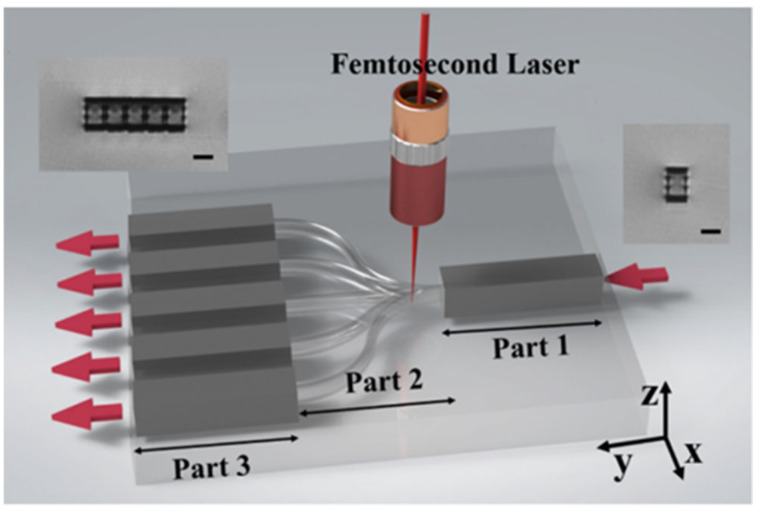
Schematic of fabrication of a hybrid-integrated beam splitter in LiTaO_3_ crystal by the fs-laser direct writing [67].

**Figure 9 micromachines-15-01101-f009:**
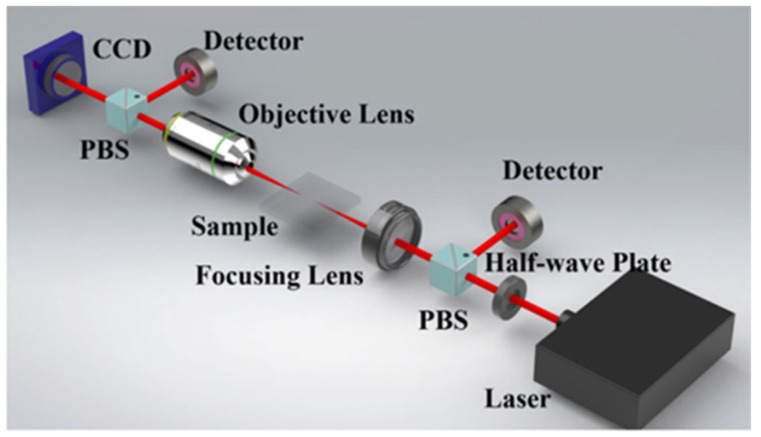
Schematic illustration of an experimental set up for a 1 × 5 beam splitter using a continuous-wave laser (central wavelength of 632.8 nm) [67].

**Figure 10 micromachines-15-01101-f010:**
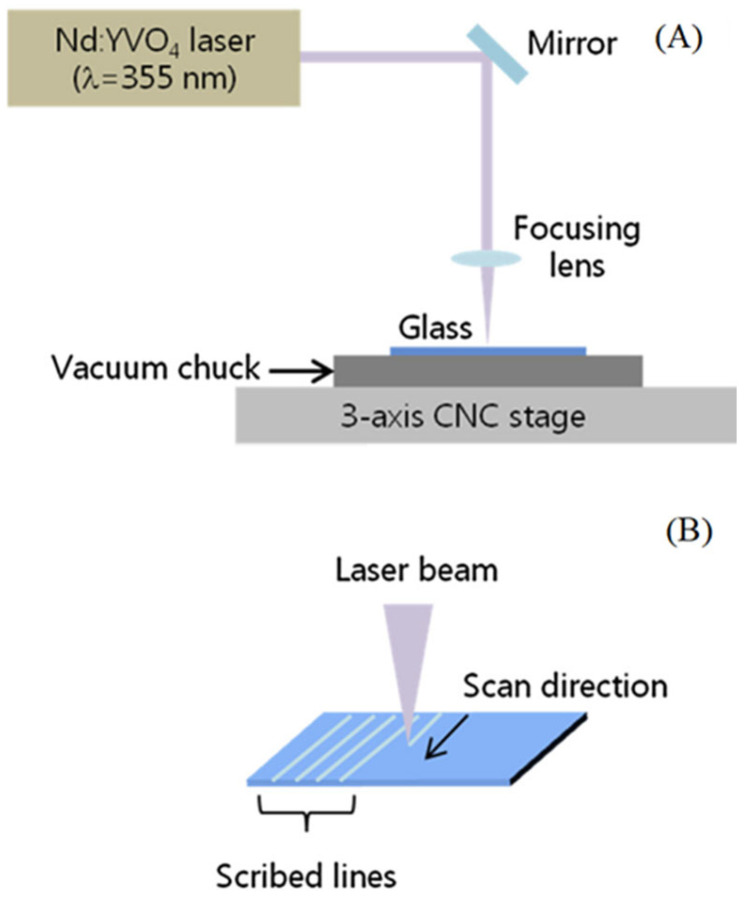
Schematic illustration of processing. (**A**) Schematic illustration of the laser scribing experimental setup. (**B**) Direction of the etch [75].

**Figure 11 micromachines-15-01101-f011:**
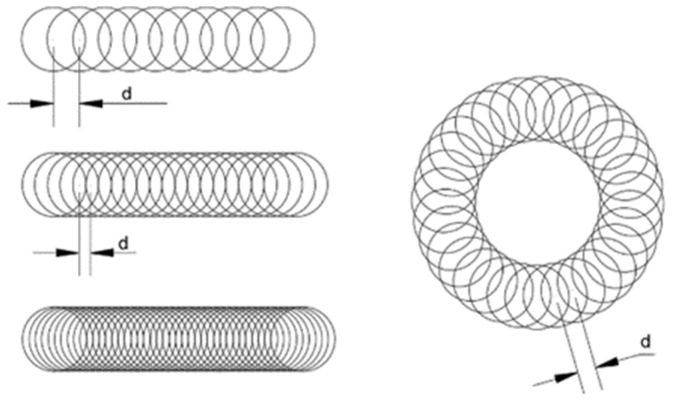
Schematic illustration of the scanning path for the helix-loop-cut approach [76].

**Figure 12 micromachines-15-01101-f012:**
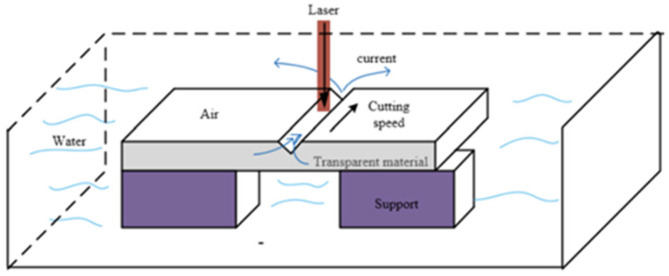
Processing schematic illustration of the water infiltration method [76].

**Table 1 micromachines-15-01101-t001:** Comparison of five processing methods: LAL, LIBWE, EALM, HBMIS, and LIPAA.

Processing Technology	Machining Interface	Primary Auxiliary Medium	Main Advantages	Main Disadvantages
LAL	Liquid–solid	HF, KOH, H_2_SO_4_	Liquids reduce the impact of hot zones, sedimentary layers, and recast layers	Liquids cause laser energy loss and degrade the beam quality
LIBWE	Liquid–solid	C_7_H_8_, C_3_H_6_O, C_1__6_H_10_, H_3_PO_4_	It is easy to remove the generated debris and achieve high processing quality	It is difficult to prepare complex 3D structures and the medium is toxic
EALM	Liquid–solid	H_2_O, C₂H_6_O	It is conducive to 3D processing and not affected by the auxiliary medium	It is difficult to etch chemically stable or corrosion-resistant materials
HBMIS	Gas–solid	Ar, He, Air	It enables layer-by-layer processing with high adaptability and flexibility	It requires the use of special equipment and processes, and the processing speed is relatively slow
LIPAA	Gas–solid	Air	No other media affecting the processed material and the processing speed is fast	Some of the laser energy is absorbed and scattered, and the hot zone is more affected

**Table 2 micromachines-15-01101-t002:** Comparison of different processing applications of ultrafast lasers for transparent materials.

Technology Application	Typical Devices	Application Direction
Micro-optics devices	Microlens arrays, raster, optical microcavities, optical waveguides, etc.	Information storage, microscopy, information sensing, etc.
Cutting and punching	Through-holes, optical trenches, special cut surfaces, etc.	Laser cutting, laser drilling, groove manufacturing, etc.
Microfluidic devices	Microfluidics passage, microfluidic chips, etc.	Drug screening, chemical microreactors, micro fuel cells, etc.
Versatile structured surface	LIPSS, black silicon, etc.	Hydrophobic materials, ultra-smooth materials and other special materials, etc.

**Table 3 micromachines-15-01101-t003:** Typical achievements of nanosecond, picosecond, and femtosecond laser processing transparent materials.

Laser Type	Application Direction	Laser Parameters	Target Material	Structural Parameters
Nanosecond pulse laser	Microfluidic devices	Laser: Nd:YVO_4_, wavelength: 1064 nm, power: 8 W, repetition rate: 10 kHz, pulse width: 20 ns	soda-lime glass	Diameter of 8 μm, depth of 1.5 μm [42]
	Cutting and punching	Laser: Nd:YAG, wavelength: 532 nm, power: 20 W, repetition rate: 10 Hz, pulse width: 12 ns	Glass (72%SiO_2_)	Depth of 2 mm [47]
Picosecond pulse laser	Microfluidic devices	Laser: DPSS laser, wavelength: 1064 nm, power: 30.6 W, repetition rate: 903 kHz, pulse width: 10 ps	SLG glass	Length of 45 mm, width of 0.45 mm [29]
	Optical waveguides	Laser: Nd:YVO_4_, wavelength: 1064 nm, power: 2 W, repetition rate: 1 MHz, pulse width: 10 ps	borosilicate and soda-lime glass	10 microns to 30 microns [83]
	Cutting and punching	wavelength: 1064 nm, power: 42 W, repetition rate: 100–1000 kHz, pulse width: 13 ps	borosilicate glass	glass thickness of 420 μm, cutting speed of 5 mm/s [56]
Femtosecond pulse laser	Optical waveguides	Laser: PHAROS-SP, wavelength: 1030 nm, pulse energy: 300–700 nJ, repetition rate: 200 kHz, pulse width: 180 fs	lanthanum borogermanate glass	Length of 9 mm [51]
	LIPSS	Laser: Wb:KGW, wavelength: 1030 nm, pulse energy: 19.1–25.9 μJ, repetition rate: 50 kHz, pulse width: 230 fs	fused silica and sapphire	Orthogonal LSFL period:932 nm, Parallel LSFL period: 965 nm [72]
	Microfluidic devices	Laser: Ti–Sapphire femtosecond laser system, wavelength: 800 nm, pulse energy: 3.5 mJ, repetition rate: 1 kHz, pulse width: 40 fs	PMMA	average surface roughness < 400 nm [84]
	Cutting and punching	Laser: Yb:KGW, wavelength: 1030 nm, power: 4.7 W, repetition rate: 25–100 kHz, pulse width: 280 fs	soda-lime glass and fused silica glass	Depth of 1 mm, aspect ratio of 20 [85]

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
