# Peer review of "Microscopic Processing of Transparent Material with Nanosecond and Ultrafast Lasers"

_micromachines, 2024, doi:10.3390/mi15091101_

Round 1

Reviewer 1 Report

Comments and Suggestions for Authors

The manuscript is a review of pulsed laser microfabrication of transparent materials. This field is very important because its applications are very extensive. The content of the manuscript includes principle of laser processing, transparent material processing with nanosecond and ultrafast lasers. The manuscript is rich in content and has reference value. However, there are some problems as below:

1.  The abstract should be written more specifically, including the main points and conclusions of the text.

2. In the part of mechanism, Fig.1 was not well interpreted in the text. The process and principle of the interaction between pulsed laser and materials are best demonstrated through an intuitive flowchart or figure. Suggest adding some key theoretical formulas for the interaction between pulsed lasers and transparent materials to explain this process. The main parameters of the interaction between pulsed laser and transparent materials should be specifically discussed.

3. What is the main conclusion of Fig.2? It should be introduced in the text. 

4. It is suggested to separately introduce the typical optical path diagram of pulse laser transparent material microfabrication in the earlier part of the manuscript, discuss the similarities and differences of nanosecond, picosecond, and femtosecond pulse laser optical path diagrams, and summarize and compare the parameters of the devices used in each.

5.Suggest adding a table to summarize the main achievements in the micro processing of transparent materials using pulsed lasers, including the materials and related parameters processed, the types and parameters of pulsed lasers used, the structures and parameters obtained, references, etc.

Author Response

Dear reviewer, thank you very much for reviewing my paper in your busy schedule and providing me with valuable comments. At the same time, thank you for giving me the opportunity to revise the article. Your comments are very important to me. I deal with every comment seriously. According to your suggestions, my reply is as follows.

Comments 1:

1.  The abstract should be written more specifically, including the main points and conclusions of the text.

Response 1: Thank you very much for raising this question. Our original abstract is too simple. Now we have added the specific introduction of each chapter according to the specific content of the article. For example, the mechanism part is introduced according to the time scale, the core argument of the nanosecond laser processing part is various processing technology means, and the core argument of the ultrafast laser processing part is the application direction of various device preparation. These have been reflected in the new abstract. You can find it from line 16 to line 21 on the first page of the revised manuscript. They are underlined in red.

Comments 2:

2. In the part of mechanism, Fig.1 was not well interpreted in the text. The process and principle of the interaction between pulsed laser and materials are best demonstrated through an intuitive flowchart or figure. Suggest adding some key theoretical formulas for the interaction between pulsed lasers and transparent materials to explain this process. The main parameters of the interaction between pulsed laser and transparent materials should be specifically discussed.

Response 2: We agree with you. In the original manuscript, figure 1 is weakly related to our introduction. This is our mistake. We did not correctly introduce it. In the revised manuscript, we re introduced the evolution of laser energy in matter according to the content in the picture and the time scale, including the conditions of electron excitation, the process of electron energy transmission, and the causes of heat accumulation. You can find them in line 76-line 100 on page 2-3. For your suggestion to add the schematic diagram of the interaction between pulsed laser and matter, we agree that it can make other readers understand this abstract process more quickly. Therefore, we added a brief schematic diagram. In revised manuscript, the picture code is Fig 2. You can find it in line 112 on page 3. We agree that you add a formula to supplement the explanation. The basic theory needs the support of the theoretical formula. At the same time, we have explained all the variables in the formula. You can find our derivation and explanation of the interaction process between laser and matter using the formula on page 3-5 line114 to line 169. Based on the one-dimensional two temperature model, we describe the process of electron and lattice temperature changes, and introduce the material ablation conditions under femtosecond pulse laser and nanosecond pulse laser respectively. All the above changes are underlined in red in the revised manuscript.

Comments 3:

3. What is the main conclusion of Fig.2? It should be introduced in the text. 

Response 3: In the original manuscript, figure 2 is to describe the relationship between processing efficiency and processing quality of nanosecond pulse laser, picosecond pulse laser and femtosecond pulse laser. We want to emphasize that in general laser processing, in terms of processing efficiency, nanosecond pulse laser>picosecond pulse laser>femtosecond pulse laser, and in terms of processing quality, nanosecond pulse laser<picosecond pulse laser<femtosecond pulse laser. But this core content is consistent with the text introduction of our previous manuscript. In order to avoid repetition, we have not described the original figure 2 more before. But what is embarrassing is that we can't get in touch with the original author of this picture, so we can't get the picture copyright through the publisher of this picture. Therefore, in the revised manuscript, we deleted this figure, but retained the description of the relationship between processing efficiency and processing quality of nanosecond pulse laser, picosecond pulse laser and femtosecond pulse laser. This section corresponds to line170-line183 on page 5 of the revised manuscript, but we did not change the text description, so we did not mark it with a red underline.

Comments 4:

4. It is suggested to separately introduce the typical optical path diagram of pulse laser transparent material microfabrication in the earlier part of the manuscript, discuss the similarities and differences of nanosecond, picosecond, and femtosecond pulse laser optical path diagrams, and summarize and compare the parameters of the devices used in each.

Response 4: We agree with your fourth suggestion. For the field of pulsed laser processing transparent materials, whether nanosecond pulsed laser, picosecond pulsed laser or femtosecond pulsed laser, the optical path will be very different according to the different conditions of light source, processing environment, auxiliary technology and so on. The thermal effect of nanosecond pulse laser is obvious during processing, so we use the classic wet etching optical path diagram as a representative. Although picosecond pulse laser can suppress the thermal effect well, with the increase of processing time, there will also be obvious thermal accumulation. In terms of reducing the effect of heat accumulation, picosecond processing also usually uses wet etching method and water film method. In order to show the difference from nanosecond laser processing, we use the light path diagram of high-pressure gas flow method as a representative. The precision of femtosecond laser processing is very high, and it is widely used in the processing of transparent materials. In recent years, with the continuous improvement of the theory, transparent materials shaped by femtosecond laser beam have received extensive attention, so we choose the optical path diagram with beam shaping as the representative. At the same time, according to the important parameters in the actual processing, we briefly described and compared the three important parameters of nanosecond pulse laser, picosecond pulse laser and femtosecond pulse laser, including the laser single pulse energy, laser repetition rate and the usual processing scale. You can find the relevant changes in line184-line222 on page 5-6 of revised manuscript. Similarly, They are underlined in red.

Comments 5:

5.Suggest adding a table to summarize the main achievements in the micro processing of transparent materials using pulsed lasers, including the materials and related parameters processed, the types and parameters of pulsed lasers used, the structures and parameters obtained, references, etc.

Response 5: We agree with you that adding a table of the results of various lasers in the micro processing of transparent materials can make readers more clearly aware of the current research progress of pulsed laser processing of transparent materials. According to the laser with different pulse width, corresponding to its main application fields, we give several examples of the processing results of transparent materials. The table includes laser type, application direction, main parameters of laser (laser type, laser wavelength, power/single pulse energy, repetition rate, pulse width), corresponding target material, basic parameters or processing parameters of processed structure. You can find it in line563-576 on page 16-17 of the revised manuscript, They are underlined in red.

  1. Additional clarifications

I'm very sorry. The Fig.9 of the original manuscript has been removed from the revised manuscript due to copyright reasons

Reviewer 2 Report

Comments and Suggestions for Authors

The paper reviews the transparent material processing by using the nanosecond and femtosecond lasers. The principles of laser material interactions, processing characteristics, advantages and disadvantages, various developed techniques to improve processing quality, and potential applications for both nanosecond and femtosecond lasers have been discussed.

The paper is well-structured and comprehensive in review. While appreciated the efforts taken, it could be more beneficial if the authors could summarize a thorough performance metrics comparison of different laser types on transparent material processing. It could be more enhanced if the authors could provide some original insights, or propose novel approaches for future research directions regarding the limitations (such as reduce thermal effects, improve efficiency, and quality) of both nanosecond and ultrafast laser processing techniques.

In general, the review paper is well-studied and provides a thorough overview of laser processing techniques for transparent materials.

Comments on the Quality of English Language

The language is formal and appropriate for a literature review, with clear and precise demonstration and statement.

Author Response

Dear reviewer, thank you very much for reviewing my paper in your busy schedule and providing me with valuable comments. At the same time, thank you for giving me the opportunity to revise the article. Your comments are very important to me. I deal with every comment seriously. According to your suggestions, my reply is as follows.

Comments 1:

1.  The abstract should be written more specifically, including the main points and conclusions of the text. The paper is well-structured and comprehensive in review. While appreciated the efforts taken, it could be more beneficial if the authors could summarize a thorough performance metrics comparison of different laser types on transparent material processing. It could be more enhanced if the authors could provide some original insights, or propose novel approaches for future research directions regarding the limitations (such as reduce thermal effects, improve efficiency, and quality) of both nanosecond and ultrafast laser processing techniques.

Response 1: Thank you very much for raising this question. We think it is a very good suggestion to compare the performance indexes of different types of lasers in the processing of transparent materials. Pulsed laser processing of transparent materials is a very important and extensive topic. According to different types of lasers, different processing devices, and different auxiliary technologies, the processing performance indicators are very diverse. In the actual processing, the pulse frequency, single pulse energy and other parameters of laser with different pulse width play an important role. At the same time, according to the different processing devices, the required laser type, laser wavelength and other parameters are also very important. Combined with the comments given by reviewer1, we briefly summarized the performance indicators of the laser in the tables line191-line222 on pages 5-6 and line567-line568 on pages 16-17 of the revised manuscript. They are underlined in red.

We agree with your suggestion on the limitations of nanosecond pulse laser and ultrafast pulse laser. Thermal effect is an unavoidable topic for nanosecond pulsed laser. At present, for the direction of nanosecond pulse laser processing transparent materials, the main solution is to change the processing environment and scanning path in the processing process to reduce the thermal stress, and carry out thermal reflux or electrochemical technology to improve the processing quality after processing. At present, many new researches show that the distribution of laser pulse energy in time can be controlled by time domain shaping. The traditional Gaussian laser can also be transformed into various interesting spatial distribution forms, such as Airy beam, Bessel beam, flat topped beam, etc., by shaping the laser pulse in spatial domain. These new technologies may be able to change the problem of nanosecond laser energy deposition. Ultra fast laser has high machining accuracy, but its machining efficiency is low, and it is difficult to process complex three-dimensional structures. At present, there are many interesting new technologies that may enable ultrafast laser processing of transparent materials to obtain better results. We have a brief summary on page 17 of revised manuscript, line568-line579 and line591-line597. They are underlined in red.

  1. Additional clarifications

I'm very sorry. The Fig.9 of the original manuscript has been removed from the revised manuscript due to copyright reasons

Round 2

Reviewer 1 Report

Comments and Suggestions for Authors

According to previous comments, the new version has made significant improvements. I think it is suitable for publication. Additionally, some figures and their annotations are too small.